# Sarcopenia and Myosteatosis Are Associated with Neutrophil to Lymphocyte Ratio but Not Glasgow Prognostic Score in Colorectal Cancer Patients

**DOI:** 10.3390/jcm11092656

**Published:** 2022-05-09

**Authors:** Raila Aro, Sanna Meriläinen, Päivi Sirniö, Juha P. Väyrynen, Vesa-Matti Pohjanen, Karl-Heinz Herzig, Tero T. Rautio, Elisa Mäkäräinen, Reetta Häivälä, Kai Klintrup, Markus J. Mäkinen, Juha Saarnio, Anne Tuomisto

**Affiliations:** 1Department of Surgery, Oulu University Hospital, 90014 Oulu, Finland; sanna.merilainen@ppshp.fi (S.M.); tero.rautio@oulu.fi (T.T.R.); elisa.makarainen-uhlback@ppshp.fi (E.M.); reetta.haivala@ppshp.fi (R.H.); kai.klintrup@oulu.fi (K.K.); juha.saarnio@oulu.fi (J.S.); 2Division of Operative Care, Medical Research Center, Oulu University Hospital, University of Oulu, 90014 Oulu, Finland; 3Cancer and Translational Medicine Research Unit, University of Oulu, P.O. Box 5000, 90014 Oulu, Finland; paivi.sirnio@oulu.fi (P.S.); juha.vayrynen@oulu.fi (J.P.V.); vesa-matti.pohjanen@oulu.fi (V.-M.P.); markus.makinen@oulu.fi (M.J.M.); 4Oulu University Hospital, Medical Research Center Oulu, P.O. Box 21, 90029 Oulu, Finland; karl-heinz.herzig@oulu.fi; 5Research Unit of Biomedicine, Department of Physiology, University of Oulu, P.O. Box 5000, 90014 Oulu, Finland; 6Department of Pediatric Gastroenterology and Metabolism, Poznan University of Medical Sciences, 61-701 Poznan, Poland

**Keywords:** sarcopenia, myosteatosis, cytokines, systemic inflammation, NLR

## Abstract

Cancer patients commonly present sarcopenia, myosteatosis, and systemic inflammation, which are risk factors of poor survival. In this study, sarcopenia and myosteatosis were defined from preoperative body computed tomography scans of 222 colorectal cancer (CRC) patients and analyzed in relation to tumor and patient characteristics, markers of systemic inflammation (modified Glasgow prognostic score (mGPS), neutrophil–lymphocyte ratio (NLR), serum levels of C-reactive protein (CRP), albumin, and 13 cytokines, and survival. Of the systemic inflammation markers, sarcopenia and/or myosteatosis associated with elevated NLR (*p* = 0.005) and low albumin levels (≤35 g/L) (*p* = 0.018), but not with mGPS or serum cytokine levels. In addition, myosteatosis was associated with a proximal tumor location (*p* = 0.039), serrated tumor subtype (*p* < 0.001), and severe comorbidities (*p* = 0.004). Multivariable analyses revealed that severe comorbidities and serrated histology were independent predictors of myosteatosis, and older age and elevated NLR were independent indicators of sarcopenia. Myosteatosis associated with shorter overall survival in univariable analysis (HR 1.959, 95% CI 1.24–3.10, *p* = 0.004) but not in multivariable analysis (*p* = 0.075). We conclude that sarcopenia and myosteatosis were associated with inflammatory marker NLR, but not with mGPS. Moreover, patients with serrated CRC may have an increased risk of myosteatosis. Myosteatosis or sarcopenia were not independent predictors of patient survival.

## 1. Introduction

Sarcopenia and myostetosis are common manifestation of frailty, and they are associated with old age and several chronic devastating illnesses, including cancer. In colorectal carcinoma patients, sarcopenia and myosteatosis precede and culminate in cancer cachexia [1,2,3,4], which is a syndrome defined by the loss of muscle mass with or without reduction in fat mass [5,6]. Sarcopenia is characterized by loss of muscle strength, function, and mass, whereas myosteatosis is defined as increased fat infiltration in muscle. Sarcopenia has been associated with older age, nutritional deficiencies, cancer, and heart and renal failure, potentially reflecting decreasing physical activity, metabolic disturbance, inflammation, and abnormal hormonal and cytokine levels [7,8,9]. Myosteatosis has been linked with aging, insulin resistance, glucocorticoid stimulation, altered leptin signaling, estrogen and androgen deficiency, and physical inactivity [10]. Sarcopenia and myosteatosis have been reported to represent risk factors for poor survival and postoperative complications in colorectal cancer (CRC) [1,11,12,13,14].

Systemic inflammation is a condition characterized by increased production of proinflammatory cytokines and acute phase proteins [15]. Systemic inflammation can be observed in 21–41% of CRC patients before surgery, most commonly in patients with an advanced disease [16,17,18]. Importantly, systemic inflammation is an independent poor prognostic factor in CRC and in other malignancies [17,18]. Frequently used biomarkers of systemic inflammation include circulating C-reactive protein (CRP) levels [19], modified Glasgow prognostic score (mGPS), a measure based on elevated serum CRP level and decreased serum albumin level [15,17,20], and blood neutrophil/lymphocyte ratio NLR [21].

In CRC, systemic inflammation has been reported to be associated with sarcopenia and myosteatosis [22,23,24], but associations with more detailed serum biomarker networks have not been conducted [25]. In this study, we aimed to investigate the relationships of sarcopenia and myosteatosis with several important circulating inflammatory mediators. Furthermore, we assessed the prognostic significance of sarcopenia and myosteatosis.

## 2. Materials and Methods

### 2.1. Data Collection

This retrospective study is based on a cohort of 378 colorectal cancer patients operated on in Oulu University Hospital during 2006–2014 who had signed written informed consent for study participation [16,26]. Patients were selected for the study if (a) preoperative venous-phase tomography was performed; (b) serum samples were taken; (c) no preoperative chemotherapy or chemoradiotherapy was given; and (d) no synchronous malignancy was observed. After application of these exclusion criteria, 222 patients were eligible.

Patient data were collected from patient registries at Oulu University Hospital, including age, comorbidities, medication, gender, tumor location, 90-day and 5-year postoperative mortality, the American Society of Anesthesiologists (ASA) class, and date of cancer recurrence. Data were collected until the patients’ death or until the end of observation period (12/2019) when the 5-year follow-up was reached for all patients. Preoperative laboratory markers including blood leucocyte differential count, CRP, hemoglobin, albumin, and carcinoembryonic antigen (CEA) were measured [16,27]. As previously described, mGPS was derived from CRP and albumin levels [20]. NLR was calculated, and NLR > 3 was regarded as high [28]. Tumor data were recorded by re-evaluation of histologic sections, including TNM8 classification, WHO2010 grading, and serrated morphology. The features favoring serrated CRC included epithelial serrations, clear or eosinophilic cytoplasm, abundant cytoplasm, vesicular nuclei, distinct nucleoli, intracellular and extracellular mucin production, and absence or scarceness of necrosis [29,30]. *BRAF* V600E mutation was detected by mutation-specific VE1 immunohistochemistry [31]. Serum cytokine levels were analyzed using a multiplex assay in patients operated on between April 2006 and January 2010 [16,26]. A total of 13 cytokines (IL-1R1, IL-4, IL-6, IL-7, CXCL8, IL-9, IL-12p70, IFNG, CXCL10, CCL2, CCL4, CCL11, and PDGF-BB) with less than three values (1.5%) outside the assay working range were included in this study.

### 2.2. Body Composition Measurements

Skeletal muscle areas were measured in the preoperative venous-phase using computed tomography (CT) examinations. A single image in which both transverse processes were visible at the level of third lumbar vertebra was selected from CT pictures. The total cross-sectional muscle area (CSMA) of abdominal (transverse, external oblique, internal oblique, and rectus abdominis) and back muscles (psoas and paraspinal muscles) in the L3 level were measured by using the range −29 to +150 Hounsfield units (HU). CSMA was normalized for patient stature and designated as skeletal muscle index. The SMI was calculated as CSMA/height^2^ [32] and BMI using the formula weight (kg)/height^2^ (m^2^). Sarcopenia was defined as SMI < 41 cm^2^/m^2^ for women and 43 cm^2^/m^2^ for men with BMI < 25 kg/m^2^ and 53 cm^2^/m^2^ for men with BMI ≥ 25 kg/m^2^ [32]. Muscle density was measured as mean HU at this cross-sectional muscle area. Myosteatosis was defined as HU < 41 for patients with BMI < 25 kg/m^2^ and HU < 33 for patients with BMI ≥ 25 kg/m^2^ [32].

### 2.3. Statistical Analysis

All analyses were performed using SPSS for Windows (IBM Corp. Released 2018. IBM SPSS Statistics for Windows, Version 27.0. Armonk, NY, USA: IBM Corp.). Pearson’s χ^2^-test or Fisher’s exact test were used for the comparison of the categorical variables. The statistical significances of the associations between categorical and continuous variables were analyzed using the Mann–Whitney test (comparing two classes) or Kruskal–Wallis test (comparing three or more classes). Multivariable binary logistic regression models of the associations of sarcopenia/myosteatosis with systemic inflammatory markers and selected clinicopathological factors using enter method were conducted. Survival curves were plotted according to the Kaplan–Meier method. Backward conditional stepwise Cox regression was used for multivariable survival modeling. Statistically significant variables from univariable analyses were considered for inclusion in multivariable analyses. Two-tailed values were considered statistically significant at *p* < 0.05.

## 3. Results

### 3.1. Patients Characteristics

We analyzed sarcopenia and myosteatosis in 222 CRC patients. The characteristics of all patients are presented in Table 1. The majority of patients were 70 or older (57%), male (53%), overweight or obese (66%), had conventional CRC (70%), and had undergone an operation with curative intent (87%). Cancer recurrence at the 5-year follow-up was diagnosed in 40 (18%) patients, and 147 (66%) patients were alive 5 years after the operation. The prevalence of sarcopenia in this cohort was 55% and the prevalence of myosteatosis was 30%. Sarcopenia and myosteatosis coexisted in 47 (21%) patients, 76 (34%) patients had sarcopenia alone, and 20 (9%) had myosteatosis alone (Figure 1a). Sarcopenia was prevalent in 47 (70%) patients with myosteatosis, whereas only 38% of patients with sarcopenia were also affected by myosteatosis. A total of 45 (20%) patients had mGPS > 0 and 55 (25%) patients had NLR > 3, but only 17 (7.7%) patients presented concurrent NLR > 3 and mGPS > 0 (Figure 1b). Hence, although both mGPS and NLR are markers of systemic inflammation, in only a few patients was systemic inflammation detected by both of these markers, and in most patients, systemic inflammation could only be detected with one of these two markers. Therefore, mGPS and NLR do not entirely mirror the same tumor–host interactions in CRC patients. 

### 3.2. Sarcopenia and Myosteatosis in Relation to Clinicopathological Characteristics 

The relationships between sarcopenia and myosteatosis and the clinicopathological variables are presented in Table 2. Sarcopenia, myosteatosis, and their combined presence associated with older age (*p* < 0.001) and higher ASA score (*p* = 0.004). Sarcopenia and concurrent sarcopenia and myosteatosis were associated with lower BMI (*p* < 0.001). Myosteatosis was associated with proximal tumor location (*p* = 0.039), serrated morphology (*p* < 0.001), and blood pressure lowering medication (*p* = 0.043).

### 3.3. Sarcopenia and Myosteatosis in Relation to Systemic Inflammatory Markers 

As our main analysis, we analyzed the relationships of sarcopenia and myosteatosis with systemic inflammatory markers. First, we evaluated the correlations between two-tiered sarcopenia and myosteatosis and systemic inflammatory markers (Appendix A and Figure 2). Sarcopenia showed a trend of a negative association with serum Hb level, and myosteatosis negatively associated with IL9, had a trend of negative association with INFg, and a trend of positive association with NLR. With the four-tiered sarcopenia–myosteatosis combination variable, NLR was elevated both in sarcopenic and myosteatotic patients, most frequently in patients with concurrent sarcopenia and myosteatosis (Table 2, *p* = 0.005). Muscle abnormalities neither associated with mGPS classification (*p* = 0.331) nor elevated CRP level (CRP ≤ 10 vs. CRP > 10, *p* = 0.584), but all patients with hypoalbuminemia (<35 g/L) had either sarcopenia or myosteatosis or both (*p* = 0.018). In addition, myosteatosis was associated with anemia (Table 3, *p* = 0.011). Comparison of serum cytokine and chemokine levels and blood cell counts in relation to sarcopenia and myosteatosis is presented in Table 3. Serum cytokine concentrations were not significantly associated with sarcopenia or myosteatosis.

### 3.4. Multivariable Analyses

The multivariable analyses showed that myosteatosis was independently associated with serrated histology (multivariable OR 3.76, 95% CI 1.82–7.76, *p* < 0.001) and higher ASA score (multivariable OR 2.66, 95% CI 1.20–5.93, *p* = 0.017, Table 4) and sarcopenia with age (multivariable HR 3.23, 95% CI 1.65–6.31, *p* < 0.001) and elevated NLR (multivariable HR 2.34, 95% CI 1.13–8.84, *p* = 0.022, Table 5).

### 3.5. Survival Analyses 

In univariable analyses based on separate, two-tiered sarcopenia and myosteatosis variables (Figure 3 and Appendix A), Kaplan–Meier curves demonstrated that sarcopenia was not statistically significantly associated with disease-free survival (DFS) (*p* = 0.102), cancer-specific survival (CSS) (*p* = 0.643), or overall survival (*p* = 0.289). Myosteatosis was associated with shorter OS (multivariable HR 1.959, 95% CI 1.24–3.10, *p* = 0.004, Figure 3A) but not with DFS (*p* = 0.704) or CSS (*p* = 0.106). In the multivariable Cox proportional hazard regression model, adjusted for tumor stage and age, myosteatosis showed a tendency toward an association with worse OS (Table 6, *p* = 0.075).

In analyses based on the four-tiered sarcopenia–myosteatosis combination variable, Kaplan–Meier curves (Figure 3) visualized that presence of sarcopenia alone was associated with shorter DFS (*p* = 0.010), but the patients with myosteatosis alone (*p* = 0.088) or both sarcopenia and myosteatosis (*p* = 0.703) had similar DFS as the patients without the muscle abnormalities. Patients with myosteatosis alone had the worse CSS (*p* = 0.002) and OS (*p* = 0.002) than patients without muscle abnormalities (Figure 3). Additionally, the patients with both myosteatosis and sarcopenia had worse OS than patients without muscle abnormalities (*p* = 0.012). In multivariable Cox analysis with the four-tiered sarcopenia–myosteatosis variable adjusted for tumor stage and age, presence of sarcopenia alone, myosteatosis alone, or both sarcopenia and myosteatosis were not statistically significantly associated with CSS and OS (Appendix A). However, presence of sarcopenia alone (*p* = 0.029) and presence of myosteatosis alone (*p* = 0.011) also associated with worse DFS in multivariable analysis.

### 3.6. Sarcopenia and Myosteatosis in Stage I–III CRC 

Since cancer metastasis is associated with systemic inflammation [21], we conducted subgroup analyses of the association of sarcopenia and myosteatosis with clinicopathological characteristics (Appendix A), and with cytokines, chemokines, and blood cell counts (Appendix A) in stage I–III CRC patients.

In this analysis, similar to stage I–IV CRCs, sarcopenia, myosteatosis, and their coexistence associated with older age (*p* < 0.001), higher ASA score (*p* < 0.001), and use of blood pressure lowering medication (*p* = 0.023). Sarcopenia and concurrent sarcopenia and myosteatosis were associated with lower BMI (*p* < 0.001) and elevated NLR (*p* = 0.006). Myosteatosis and concurrent myosteatosis and sarcopenia were associated with proximal tumor location (*p* = 0.021), serrated morphology (*p* = 0.002), and lower albumin level (≤35 g/L). However, when metastatic CRC patients were excluded, *BRAF* mutation also associated with myosteatosis alone or together with sarcopenia (*p* = 0.045). Serum cytokine concentrations were not significantly associated with sarcopenia or myosteatosis. Only lower IL-9 level associated with presence of myosteatosis, sarcopenia, or both muscle abnormalities (*p* = 0.037). Multivariable regression models found the same factors associating with sarcopenia and myosteatosis in stage I–III patients as in stage I–IV patients: Myosteatosis associated with serrated histology (multivariable OR 3.54, 95% CI 1.56–8.00, *p* = 0.002) and higher ASA score (multivariable OR 4.42, 95% CI 1.70–11.47, *p* = 0.002, Appendix A) and sarcopenia with age (multivariable HR 3.92, 95% CI 1.87–8.23, *p* < 0.001) and elevated NLR (multivariable HR 3.17, 95% CI 1.38–7.28, *p* = 0.007, Appendix A). 

In survival analyses of stage 1–3 patients, sarcopenia did not associate with OS, CSS, or DFS, and myosteatosis associated only with shorter OS (*p* = 0.021), not with CSS or DFS (Appendix A). However, in stage I–III patients, myosteatosis also significantly associated with poor OS in the multivariable Cox proportional hazard regression model adjusted for tumor stage and age, (multivariable HR 1.81, 95% CI 1.00–3.26, *p* = 0.048, Appendix A).

## 4. Discussion

We evaluated the associations between muscle density abnormalities, systemic inflammation, and patient characteristics. We found that 55% of CRC patients were affected by sarcopenia and 30% by myosteatosis. Preoperative sarcopenia and myosteatosis associated with advanced age, lower BMI, and elevated NLR, but not with mGPS classification. Moreover, myosteatosis associated with higher ASA grade, proximal location, and serrated morphology.

Previous studies have reported a wide prevalence range of both sarcopenia (12–60%) [2,3,11,23,33,34,35,36] and myosteatosis (30–78%) [2,3,33,37] in CRC patients. A reason for this variation may be patients’ ethnicity, sex, and different skeletal muscle cutoff points. The sarcopenia and myosteatosis prevalence in our cohort are comparable to these previous studies. In our study, in 21% of the CRC patients, sarcopenia and myosteatosis occurred concurrently. To our knowledge, the effects of concurrent presence of sarcopenia and myosteatosis in CRC have only been reported in one publication [33], in which the prevalence of concurrent presence of sarcopenia and myosteatosis was similar to our study (around 20%).

Systemic inflammation is regarded as an important factor driving cachexia-related symptoms [5], and recent meta-analysis of 11,474 CRC patients indicated a consistent association between sarcopenia and systemic inflammation [38]. However, in our cohort, sarcopenia, myosteatosis, or presence of both were not significantly associated with an elevated mGPS. Dolan et al. [22] found an increased incidence of both sarcopenia and myosteatosis with mGPS 2 class, which was defined by elevated CRP levels and decreased albumin levels. In our cohort, only 4 (1.8%) patients were classified as mGPS 2, whereas in the study by Dolan et al. [22] 13.5% of the patients were classified as mGPS 2, and in a study by McMillan et al. [20], 12% of patients were mGPS 2. In our study, hypoalbuminemia was detected only in 13 (5.9%) CRC patients, and both sarcopenia and myosteatosis associated with decreased albumin levels (*p* = 0.029 and *p* = 0.003). In many other studies, the reported prevalence of hypoalbuminemia in patients with operable CRC has been higher, 22–30% [4,39]. Hence, the low hypoalbuminemia prevalence in our cohort may account for the lack of association between muscle abnormalities and mGPS in our study. In our cohort, systemic inflammation was present in 20% of CRC patients when evaluated by the mGPS classification and in 24.8% when defined by NLR. Dolan et al. [22] found a similar proportion of systemically inflamed patients with 23% of the CRC patients based on mGPS classification and 40% of CRC patients based on increased NLR.

Both mGPS and elevated NLR are regarded as markers of systemic inflammation [17,40]. Although there was an association between NLR and mGPS (*p* = 0.016, data not shown), only 31% of patients with raised NLR also had elevated mGPS, and the majority of patients with increased NLR had normal CRP and albumin levels (Figure 1b). Thus, increased NLR and elevated mGPS score may have both common and unrelated causes. CRP and albumin, the measured variables in mGPS classification, are acute phase proteins reflecting elevated circulating IL6 levels [41]. NLR is elevated as a response to inflammation, but also in response to metabolic alterations such as hyperglycemia [42] and insulin resistance [43,44]. Concordant with the lack of association between muscle abnormalities and mGPS classification, serum cytokine levels were not significantly associated with muscle abnormalities, either (Table 3).

In our study, sarcopenic and myosteatotic patients were more often over 70 years. In the literature, 4.6–36.5% of people at an average age of 70 were sarcopenic [45]. Age is a well-known risk factor for sarcopenia, and in our cohort higher age together with elevated NLR were associated with sarcopenia in multivariable analysis, whereas serrated morphology and higher ASA classification associated with myosteatosis. Therefore, sarcopenia and myosteatosis are muscle abnormalities potentially driven by different mechanisms.

Myosteatosis was associated with the serrated histological subtype of CRC (*p* < 0.001). Serrated CRCs constitute 30% of all CRCs, characteristically developing from serrated precursor lesions. One of the typical molecular features of serrated CRC is the activation of the MAPK-ERK signaling pathway via activating mutations of *BRAF* or *KRAS* [46]. In the four-tiered sarcopenia–myosteatosis variable, neither sarcopenia nor myosteatosis were significantly associated with the *BRAF* mutation, but when evaluated as separate variables, myosteatosis associated with the *BRAF* mutation (*p* = 0.024, data not shown). Mutated *KRAS* alters glucose metabolism leading to the production of ATP and other metabolic intermediates in anaerobic glycolysis, and not by citric acid synthesis [47]. Mutant *KRAS* also upregulates expression of the GLUT1 glucose transporter, promoting glucose uptake by cells. Interestingly, several studies have demonstrated higher glucose accumulation in 18F-FDG PET/CT scans in CRCs with *KRAS*/*NRAS*/*BRAF* mutations [48]. The association between myosteatosis and serrated morphology suggests that serrated CRC patients might have an increased risk of myosteatosis, and we hypothesize that this may reflect higher glucose metabolism in serrated CRC.

Myosteatosis was associated with proximal tumor location in our cohort and also in other studies [49]. However, multivariable analysis revealed that only serrated histology, not proximal location, was independently associated with myosteatosis. Serrated CRCs are most often localized in proximal colon [50] suggesting that the serrated morphology may be a stronger determinant of myosteatosis than proximal tumor location.

Several studies have indicated the association between sarcopenia and myosteatosis and DFS, CSS, and OS [22,32,51,52,53,54], but also some have not [3,12]. In our cohort, myosteatosis or sarcopenia alone or concurrent sarcopenia and myosteatosis were not associated with survival, although myosteatosis showed a tendency toward an association with worse OS. When the CRC patients with distant metastases were excluded, myosteatosis independently associated with OS. In some of the previous studies [22,32,53], sample size has been larger than in our cohort, which might influence our negative result. In particular, it is not clear why the patients with concurrent sarcopenia and myosteatosis appeared to have similar DFS as those with neither muscle abnormality. Sarcopenia and myosteatosis are multifactorial muscle abnormalities, also associated with sex/race/ethnicity. The studies by Nakanishi et al. [12] and Miyamoto et al. [54] included patients of different ethnicities and cutoff values for muscle abnormalities, which complicates comparison of results.

In addition to the relatively small sample size, some other limitations should be acknowledged. The definition of sarcopenia was limited to myopenia without measuring muscle strength, as such data were not available for this cohort. Multiple factors may contribute to myosteatosis and sarcopenia, such as the patient’s lifestyle and diet, nutritional care before and after surgery, and other environmental exposures, in addition to effects of the tumor and inflammation. While we extensively characterized tumors and systemic inflammatory markers, we did not collect diet or lifestyle data, potentially resulting in residual confounding. Muscle density and circulating biomarker data were based on single preoperative measurements, and follow-up of their development was lacking. Our analysis was retrospective, and further larger prospective studies are required to validate the findings. The strength of our study was the measurement of multiple inflammatory biomarkers, enabling more granular analyses than single biomarker measurements. As the correlations between serum cytokine concentrations and sarcopenia or myosteatosis were generally quite weak, further studies are required to evaluate the molecular mechanisms underlying sarcopenia and myosteatosis.

## 5. Conclusions

In conclusion, sarcopenia and myosteatosis were associated with elevated NLR but not with the mGPS score of systemic inflammation. Myosteatosis was also associated with the serrated CRC subtype, suggesting that patients with serrated CRC might be at higher risk of myosteatosis.

## Figures and Tables

**Figure 1 jcm-11-02656-f001:**
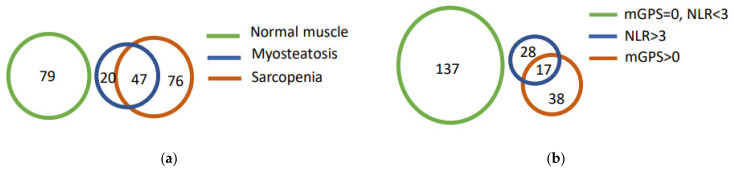
(**a**) Diagram of body composition parameters in CRC patients. In total, 47 patients had coexisting sarcopenia and myosteatosis. (**b**) Diagram of systemic inflammation markers in CRC patients. Only 17 patients had both NLR > 3 and mGPS > 0.

**Figure 2 jcm-11-02656-f002:**
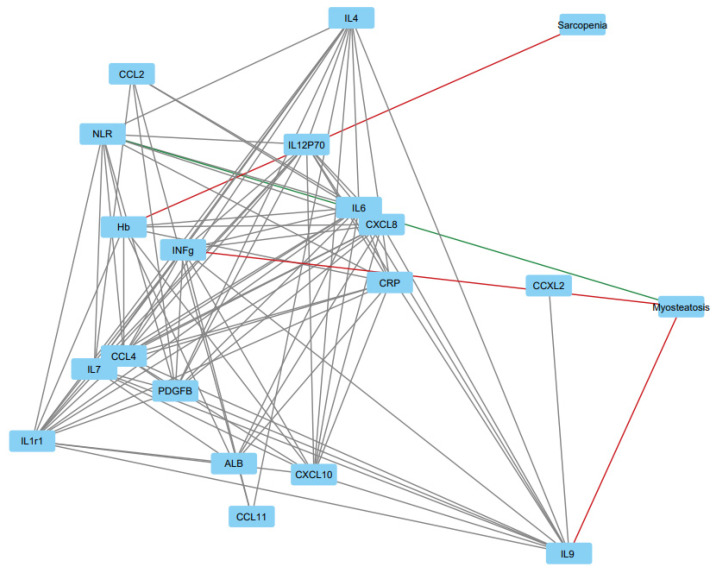
Correlation network between sarcopenia, myosteatosis, serum cytokines, CRP, Hb, and albumin established by the Cytoscape software platform utilizing the Perfuse force-directed algorithm. Variables are presented by nodes and their associations by edges. Correlations with *p* < 0.1 are shown and the edge length illustrates the significance of the association. The green edge illustrates a positive correlation between myosteatosis and NLR, and red edges negative correlations between sarcopenia/myosteatosis and other variables.

**Figure 3 jcm-11-02656-f003:**
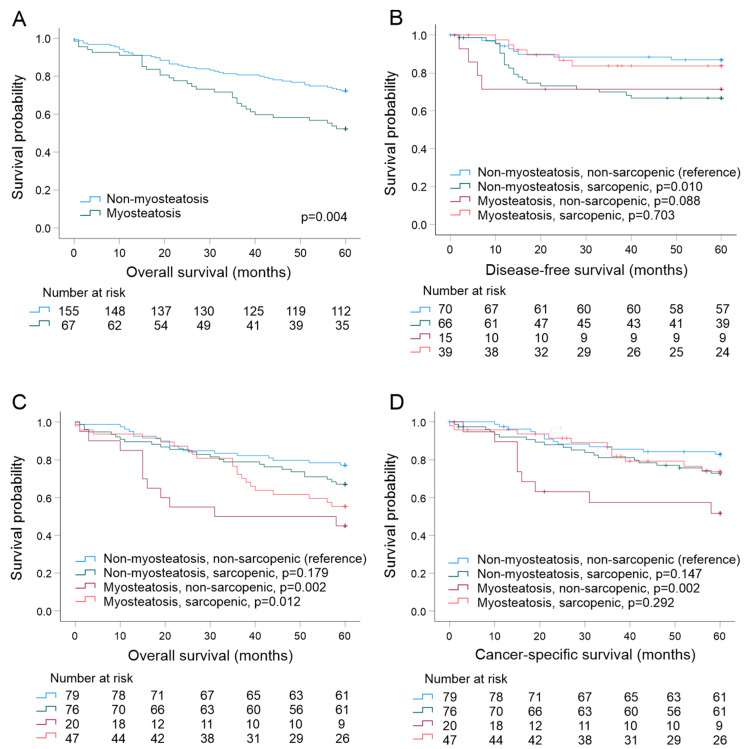
Kaplan–Meier survival curves for (**A**) the presence of myosteatosis and overall survival, (**B**) the presence of myosteatosis and sarcopenia and disease-free survival, (**C**) the presence of myosteatosis and sarcopenia and overall survival, and (**D**) the presence of myosteatosis and sarcopenia and cancer-specific survival.

**Table 1 jcm-11-02656-t001:** Characteristics of colorectal cancer patients.

Colorectal Cancer Patients (n = 222)	n (%)
Body composition	
Normal muscle	79 (35.6%)
Myosteatosis alone	20 (9.0%)
Sarcopenia alone	76 (34.2%)
Both myosteatosis and sarcopenia	47 (21.2%)
Age	
<70 years old	95 (42.8%)
≥70 years old	127 (57.2%)
Gender	
Male	117 (52.7%)
Female	105 (47.3%)
BMI categories	
Underweight < 18.5 kg/m^2^	3 (1.4%)
Normal weight 18.5–24.9 kg/m^2^	73 (32.9%)
Overweight 25–29.9 kg/m^2^	92 (41.4%)
Obese ≥ 30 kg/m^2^	54 (24.3%)
Tumor location	
Proximal colon	89 (40.1%)
Distal colon	58 (26.1%)
Rectum	75 (33.8%)
TNM stage	
Stage I	42 (18.9%)
Stage II	75 (33.8%)
Stage III	73 (32.9%)
Stage IV	32 (14.4%)
Depth of invasion	
T1	13 (5.9%)
T2	43 (19.4%)
T3	152 (68.5%)
T4	14 (6.3%)
Nodal metastases	
N0	120 (54.1%)
N1	67 (30.2%)
N2	35 (15.8%)
Distant metastases	
M0	190 (85.6%)
M1	32 (14.4%)
Morphology	
conventional	156 (70.3%)
serrated	66 (29.7%)
Operation type	
Curative	192 (86.5%)
Palliative	30 (13.5%)
ASA classification	
1 (no systemic diseases)	13 (5.9%)
2 (mild systemic disease)	96 (43.2%)
3 (severe systemic disease)	84 (37.8%)
4 (very severe systemic disease)	18 (8.1%)
Coronary artery disease	
No	177 (79.7%)
Yes	45 (20.3%)
Diabetes	
No	185 (83.3%)
Yes	37 (16.7%)
Preoperative blood samples	
Hemoglobin, g/L	125.7 (±17.0) [86–167]
CEA, μg/L	32.3 (±188.8) [0.5–2423.0]
Albumin, g/L	42.7 (±4.4) [21–69]
C-reactive protein, mg/L	10.4 (±25.3) [0.0–189.0]
NLR	
≤3	165 (75%)
>3	55 (25%)
Modified Glasgow prognostic score	
0	177 (79.7%)
1	41 (18.5%)
2	4 (1.8%)
5-year cancer recurrence	43 (19.4%)
Died in 90 days	8 (3.6%)
Died in 5 years	75 (33.8%)
Follow-up, month	82 (± 45.3) [0.10–164.5]

ASA: American Society of Anesthesiologists; BMI: body mass index; CEA: carcinoembryonic antigen; NLR: neutrophil–lymphocyte ratio.

**Table 2 jcm-11-02656-t002:** Distribution of clinocopathological factors in myosteatosis and sarcopenia patient groups.

Factors	Neither Sarcopenia Nor Myosteatosis (n = 79)	Sarcopenia Only, No Myosteatosis (n = 76)	Myosteatosis Only, No Sarcopenia (n = 20)	Both Sarcopenia and Myosteatosis (n = 47)	*p*-Value
Age, mean, years, (±SD)	65 (±11.2)	70 (±11.0)	72 (±9.9)	76 (±9.7)	<0.001
Age					
≤70 years (n = 95)	51 (64.6%)	27 (35.5%)	6 (30.0%)	11 (23.4%)	<0.001
>70 years (n = 127)	28 (35.4%)	49 (64.5%)	14 (70.0%)	36 (76.6%)	
Gender					
Male (n = 117)	48 (60.8%)	40 (52.6%)	8 (40.0%)	21 (44.7%)	0.211
Female (n = 105)	31 (39.2%)	36 (47.4%)	12 (60.0%)	26 (55.3%)	
BMI, kg/m^2^					
<18.5 (n = 3)	1 (1.3%)	2 (2.6%)	0 (0.0%)	0 (0.0%)	<0.001
18.5–24.9 (n = 73)	27 (34.2%)	15 (19.7%)	6 (30.0%)	25 (53.2%)	
25–29.9 (n = 92)	24 (30.4%)	47 (61.8%)	7 (35.0%)	14 (29.8%)	
>30 (n = 54)	27 (34.2%)	12 (15.8%)	7 (35.0%)	8 (17.0%)	
Tumor location					
Proximal colon (n = 89)	28 (35.4%)	24 (31.6%)	11 (55.0%)	26 (55.3%)	0.039
Distal colon (n = 58)	17 (21.5%)	26 (34.2%)	5 (25.0%)	10 (21.3%)	
Rectum (n = 75)	34 (43.0%)	26 (34.2%)	4 (20.0%)	11 (23.4%)	
WHO grade					
Grade 1 (n = 56)	20 (25.3%)	16 (21.1%)	6 (30.0%)	14 (29.8%)	0.176
Grade 2 (n = 140)	53 (67.1%)	49 (64.5%)	14 (70.0%)	24 (51.1%)	
Grade 3 (n = 26)	6 (7.6%)	11 (14.5%)	0 (0.0%)	9 (19.1%)	
TNM stage					
I (n = 42)	18 (22.8%)	8 (10.5%)	5 (25.0%)	11 (23.4%)	0.398
II (n = 75)	27 (34.2%)	27 (35.5%)	5 (25.0%)	16 (34.0%)	
III (n = 73)	24 (30.4%)	31 (40.8%)	5 (25.0%)	13 (27.7%)	
IV (n = 32)	10 (12.7%)	10 (13.2%)	5 (25.0%)	7 (14.9%)	
Depth of invasion					
T1 (n = 13)	7 (8.9%)	3 (3,9%)	1 (5.0%)	2 (4.3%)	0.193
T2 (n = 43)	16 (20.3%)	9 (11.8%)	6 (30.0%)	12 (25.5%)	
T3 (n = 152)	50 (63.3%)	61 (80.3%)	13 (65.0%)	28 (59.6%)	
T4 (n = 14)	6 (7.6%)	3 (3.9%)	0 (0.0%)	5 (10.6%)	
Nodal metastases					
N0 (n = 120)	46 (58.2%)	36 (47.4%)	10 (50.0%)	28 (59.6%)	0.382
N1 (n = 67)	19 (24.1%)	30 (39.5%)	5 (25.0%)	13 (27.7%)	
N2 (n = 35)	14 (17.7%)	10 (13.2%)	5 (25.0%)	6 (12.8%)	
Distant metastases					
M0 (n = 190)	69 (87.3%)	66 (86.8%)	15 (75.0%)	40 (85.1%)	0.541
M1 (n = 32)	10 (12.7%)	10 (13.2%)	5 (25.0%)	7 (14.9%)	
Serrated morphology					
yes (n = 66)	18 (22.8%)	15 (19.7%)	9 (45.0%)	24 (51.1%)	<0.001
no (n = 156)	61 (77.2%)	61 (80.3%)	11 (55.0%)	23 (48.9%)	
ASA grade					
I (n = 13)	7 (9.1%)	4 (5.4%)	1 (5.6%)	1 (2.4%)	0.004
II (n = 96)	46 (59.7%)	34 (45.9%)	4 (22.2%)	12 (28.6%)	
III (n = 84)	18 (23.4%)	31 (41.9%)	11 (61.1%)	24 (57.1%)	
IV (n = 18)	6 (7.8%)	5 (6.8%)	2 (11.1%)	5 (11.9%)	
Diabetes					
No (n = 185)	66 (83.5%)	67 (88.2%)	15 (75.0%)	37 (78.7%)	0.354
Yes (n = 37)	13 (16.6%)	9 (11.8%)	5 (25.0%)	10 (21.3%)	
Coronary artery disease					
No (n = 177)	67 (84.8%)	62 (81.6%)	13 (65.0%)	35 (74.5%)	0.177
Yes (n = 45)	12 (15.2%)	14 (18.4%)	7 (35.0%)	12 (25.5%)	
Use of blood pressure lowering medication					
No (n = 95)	43 (54.4%)	31 (40.3%)	6 (30.0%)	15 (31.9%)	0.043
Yes (n = 127)	36 (45.6%)	45 (59.2%)	14 (70.0%)	32 (68.1%)	
Use of cholesterol lowering medication					
No (n = 146)	54 (68.4%)	46 (60.5%)	11 (55.0%)	35 (74.5%)	0.290
Yes (n = 76)	25 (31.6%)	30 (39.5%)	9 (45.0%)	12 (25.5%)	
CRP, mg/L					
≤10 (n = 176)	63 (79.7%)	60 (78.9%)	18 (90.0%)	35 (74.5%)	0.584
>10 (n = 46)	16 (20.3%)	16 (21.1%)	2 (10.0%)	12 (25.5%)	
Modified Glasgow prognostic score					
0 (n = 177)	63 (79.7%)	61 (80.3%)	18 (90.0%)	35 (74.5%)	0.331
1 (n = 41)	16 (20.3%)	14 (18.4%)	2 (11.8%)	9 (19.1%)	
2 (n = 4)	0 (0.0%)	1 (1.3%)	0 (0.0%)	3 (6.4%)	
NLR					
≤3 (n = 165)	68 (87.2%)	53 (70.7%)	16 (76.5%)	28 (59.6%)	0.005
>3 (n = 55)	10 (12.8%)	22 (29.3%)	4 (23.5%)	19 (40.4%)	
Albumin, g/L					
≤35 (n = 9)	0 (0.0%)	3 (3.9%)	1 (5.0%)	5 (10.6%)	0.018
>35 (n = 213)	79 (100.0%)	73 (96.1%)	19 (95.0%)	42 (89.4%)	
BRAF VE1 immunohistochemistry					
Negative (n = 196)	72 (91.1%)	70 (92.1%)	16 (80.0%)	38 (80.9%)	0.126
Positive (n = 26)	7 (8.9%)	6 (7.9%)	4 (20.0%)	9 (19.1%)	

ASA: American Society of Anesthesiologists; BMI: body mass index; CRP: C-reactive protein; NLR: neutrophil–lymphocyte Ratio.

**Table 3 jcm-11-02656-t003:** Serum cytokine and chemokine levels, blood cell counts, and other laboratory parameters in relation to sarcopenia and myosteatosis.

Factors	Neither Sarcopenia Nor Myosteatosis ^1^	Sarcopenia Only, No Myosteatosis ^2^	Myosteatosis Only, No Sarcopenia ^3^	Both Sarcopenia and Myosteatosis ^4^	*p*-Value
Laboratory parameters, median (IQR)				data	data
Serum CRP, mg/L	2.24 (0.70–8.17)	2.00 (0.71–7.00)	2.80 (0.91–7.00)	4.00 (1.00–11.81)	0.536
Serum Albumin, g/L	43.00 (41.0–45.0)	43.0 (41.0–45.0)	43.0 (39.3–44.8)	42.0 (38.0–45.0)	0.264
Blood leukocytes ^5^	6.70 (5.60–7.90)	6.65 (5.60–7.98)	7.60 (6.48–8.70)	7.00 (5.80–8.80)	0.197
Blood neutrophils ^5^	3.80 (2.88–4.99)	4.00 (2.90–5.00)	4.70 (3.93–5.80)	4.40 (3.30–5.50)	0.102
Blood lymphocytes ^5^	1.90 (1.50–2.33)	1.80 (1.30–2.10)	1.95 (1.63–2.58)	1.70 (1.20–2.30)	0.133
Blood NLR	2.08 (1.47–2.62)	2.22 (1.60–3.26)	2.09 (1.64–2.83)	2.67 (1.71–3.43)	0.017
Blood monocytes ^5^	0.60 (0.46–0.70)	0.60 (0.40–0.74)	0.70 (0.53–0.80)	0.70 (0.50–0.80)	0.024
Hemoglobin, g/L	130 (118–142)	128 (110–139)	123 (111–140)	120 (109–129)	0.011
CEA, μg/L	1.60 (1.00–3.98)	1.80 (1.10–7.90)	2.85 (1.00–20.68)	2.70 (1.40–6.73)	0.104
Cytokines, pg/mL, median (IQR)					
IL-1R1	80.8 (50.8)	61.9 (58.8)	46.6 (68.7)	50.7 (83.5)	0.664
IL-4	1.00 (0.37)	0.86 (0.50)	0.78 (0.28)	0.87 (0.41)	0.336
IL-6	6.47 (5.96)	4.50 (4.51)	4.58 (3.46)	4.69 (10.20)	0.474
IL-7	6.24 (3.35)	5.34 (3.61)	5.01 (2.24)	4.84 (4.00)	0.577
CXCL8	13.4 (6.90)	10.9 (8.03)	11.5 (3.39)	11.3 (10.34)	0.713
IL-9	9.62 (7.27)	7.57 (10.70)	6.88 (10.13)	5.65 (8.04)	0.106
IL-12	30.4 (25.6)	31.6 (24.1)	31.7 (24.2)	27.9 (30.5)	0.684
IFNg	37.7 (17.4)	30.8 (22.9)	25.3 (16.9)	27.3 (20.6)	0.104
CXCL10	858 (489)	924 (539)	1068 (1055)	973 (738)	0.763
CCL2	14.6 (9.3)	15.4 (20.5)	17.5 (11.3)	14.7 (17.1)	0.639
CCL4	64.5 (35.7)	59.4 (18.6)	58.6 (59.1)	69.0 (39.8)	0.550
CCL11	138 (67.6)	123 (83.8)	155 (96.7)	114 (77.7)	0.492
PDGF-BB	9280 (4823)	8412 (5360)	8935 (7759)	8577 (10810)	0.986

CRP: C-reactive protein; NLR: neutrophil–lymphocyte ratio; CEA: carcinoembryonic antigen; ^1^: N = 79 in laboratory parameters, N = 30 in cytokine measurements; ^2^: N = 36 in laboratory parameters, N = 36 in cytokine measurements; ^3^: N = 20 in laboratory parameters, N = 7 in cytokine measurements; ^4^: N = 47 in laboratory parameters, N = 16 in cytokine measurements; ^5^: ×10^9^/L.

**Table 4 jcm-11-02656-t004:** Multivariable binary regression model for myosteatosis probability.

Factors	OR	95% CI	*p*-Value
Age (<70 vs. ≥70 years)	1.92	0.87–4.23	0.105
Tumor location (proximal vs. distal colon)	0.583	0.24–1.40	0.231
Tumor location (proximal colon vs. rectum)	0.61	0.27–1.36	0.228
Serrated morphology (yes vs. no)	3.76	1.82–7.76	<0.001
ASA grade (I–II vs. III–IV)	2.66	1.20–5.93	0.017
NLR (<3 vs. >3)	1.39	0.64–2.96	0.407
Albumin level (<35 g/L vs. >35 g/L)	0.32	0.07–1.51	0.153
Blood pressure lowering medication (yes/no)	1.15	0.53–2.50	0.730

Abbreviations: OR: odds ratio; CI: confidence interval; ASA: American Society of Anesthesiologists; NLR: neutrophil–lymphocyte ratio.

**Table 5 jcm-11-02656-t005:** Multivariable binary regression model for sarcopenia probability.

Factors	OR	95% CI	*p*-Value
Age (<70 vs. ≥70 years)	3.23	1.65–6.31	<0.001
Tumor location (proximal vs. distal colon)	1.78	0.80–3.95	0.157
Tumor location (proximal colon vs. rectum)	1.12	0.55–2.28	0.751
Serrated morphology (yes vs. no)	1.13	0.58–2.21	0.723
ASA grade (I–II vs. III–IV)	1.07	0.52–2.19	0.860
NLR (<3 vs. >3)	2.34	1.13–4.84	0.022
Albumin level (<35 g/L vs. >35 g/L)	0.22	0.03–2.02	0.181
Blood pressure lowering medication (yes/no)	0.84	0.43–1.65	0.612

Abbreviations: OR: odds ratio; CI: confidence interval; ASA: American Society of Anesthesiologists; NLR: neutrophil–lymphocyte ratio.

**Table 6 jcm-11-02656-t006:** Multivariable Cox regression model for overall survival.

Factors	HR	95% CI	*p*-Value
Age (<70 vs. ≥70)	2.69	1.58–4.60	<0.001
Tumor invasion (T1–T2 vs. T3–T4)	1.22	0.68–2.19	0.497
Nodal metastases (N0 vs. N1–N2)	2.74	1.60–4.68	<0.001
Distant metastases (M0 vs. M1)	5.54	3.19–9.62	<0.001
Myosteatosis (No vs. Yes)	1.55	0.96–2.50	0.075

Abbreviations: HR: hazard ratio; CI: confidence interval. Median follow-up time 60 months; 75 (33.8%) events.

## Data Availability

Not applicable.

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
