# Peer review of "Sarcopenia and Myosteatosis Are Associated with Neutrophil to Lymphocyte Ratio but Not Glasgow Prognostic Score in Colorectal Cancer Patients"

_jcm, 2022, doi:10.3390/jcm11092656_

Round 1

Reviewer 1 Report

The authors showed that myosteatosis associates with NLR, a systemic inflammation marker, but not GPS. It also associates with serrated CRC but  does not correlate with patient survival. The message is clear but there are some points to be clarified.

Major points:

  1. Explanation of how serrated morphology was diagnosed should be described. The "serrated cancer" is more molecular and genetic diagnosis (i.e. MSI, BRAF, etc.) than morphology according to WHO category. How did this diagnosed, and is the frequency in this study (29.7%) consistent with other studies? 
  2. Please explain the results of multivariate analysis in Table 4 & 5 in more detail. How did you adjusted? Were the factors included in the analysis all showed in the table? If so, why the other factors such as blood pressure lowering medication, which should be a significant factor in univariate analysis, are excluded? If the analysis method is not justified, the results sound arbitrary.
  3. Fig 1b. Is this figure necessary for this paper? There is no discussion and also it seems out of aim.
  4. Fig 2. The description "green edge illustrates a positive correlation between myosteatosis and CXCL10". Is this correct? The network seems myosteatosis is correlating with CXCL8. Am I wrong? In addition, there is no discussion or assessment about this network. If this figure does not show any significance, I would recommend to move to supplementary figure.

Minor points:

  1. Line 95-97. The sentence sounds strange. Please check.
  2. Table 1. Lack of % in morphology.

Author Response

Reviewer 1: Comments and Suggestions for Authors

The authors showed that myosteatosis associates with NLR, a systemic inflammation marker, but not GPS. It also associates with serrated CRC but does not correlate with patient survival. The message is clear but there are some points to be clarified.

Major points:

  1. Explanation of how serrated morphology was diagnosed should be described. The "serrated cancer" is more molecular and genetic diagnosis (i.e. MSI, BRAF, etc.) than morphology according to WHO category. How did this diagnosed, and is the frequency in this study (29.7%) consistent with other studies?

Response: In this study, serrated morphology was defined by established histological criteria that are corporated in WHO classification of tumours (Tuppurainen et al., 2005, WHO2010) We have now clarified this in materials and methods: The features favoring serrated CRC included epithelial serrations, clear or eosinophilic cytoplasm, abundant cytoplasm, vesicular nuclei, distinct nucleoli, intracellular and extracellular mucin production, and absence or scarceness of necrosis (Tuppurainen et al., 2005, WHO2010).

We agree that many serrated CRCs can be detected based on molecular features. However, even with several useful markers (such as microsatellite instability and BRAF mutation) there is no unequivocal molecular definition for serrated CRC. The percentage of colorectal cancers that are classified as serrated carcinomas vary between 10-30% according to the criteria used in their detection, and several reviews approximate that around one third of CRCs arise via serrated pathway (Snover, 2011). Therefore, the frequency of serrated CRCs in our cohort is concordant with earlier reports.

  1. Please explain the results of multivariate analysis in Table 4 & 5 in more detail. How did you adjusted? Were the factors included in the analysis all showed in the table? If so, why the other factors such as blood pressure lowering medication, which should be a significant factor in univariate analysis, are excluded? If the analysis method is not justified, the results sound arbitrary.

Response: In the analyses presented in the original version of the manuscript, the factors associating with myosteatosis in univariate analysis (Table 2) were included in Table 4 to find independent predictors of myosteatosis, and factors associating with sarcopenia in univariate analysis (Table 2) were included in Table 5. However, we agree that the selection of factors may have seemed arbitrary. Hence, we have now clarified the selection criteria and re-performed these analyses using all statistically significant variables from Table 2 included in both multivariable models (Tables 4 and 5). Tables 4 and 5 have been modified accordingly. The main results remained the same: Myosteatosis independently associated with serrated morphology and ASA score, and sarcopenia with age and elevated NLR.

  1. Fig 1b. Is this figure necessary for this paper? There is no discussion and also it seems out of aim.

Response: Figure 1b well illustrates that although the Glasgow Prognostic score and blood Neutrophil/lymphocyte ratio are both markers of systemic inflammation, in only 17 CRCs (7.7%) systemic inflammation is indicated by both markers, whereas systemic inflammation is detected in 45 CRCs (20.5%) by NLR alone and in 55 CRCs (25.0%) by mGPS alone. Hence, mGPS and NLR do not mirror exactly same tumor-host interactions in CRC patients. This information is needed to clarify why sarcopenia associates with systemic inflammation identified by NLR but not with systemic inflammation identified by mGPS. We now refer to Figure 1b in the discussion, line 291-293.

  1. Fig 2. The description "green edge illustrates a positive correlation between myosteatosis and CXCL10". Is this correct? The network seems myosteatosis is correlating with CXCL8. Am I wrong? In addition, there is no discussion or assessment about this network. If this figure does not show any significance, I would recommend to move to supplementary figure.

Response: We thank the reviewer for the careful review of Figure 2. Indeed, the only positive association illustrated in Figure 2 exists between myosteatosis and NLR (Supplementary Table 1), not between myosteatosis and CXCL8/CXCL10. We have corrected the Figure 2 caption accordingly. The findings of this correlation plot (and Supplementary table 1) are depicted immediately after the reference in the text line 156-160.

First, we evaluated the correlations between two-tiered sarcopenia and myosteatosis and systemic inflammatory markers (Supplementary Table 1, Figure 2). Sarcopenia showed a trend of a negative association with serum Hb level, and myosteatosis negatively associated with IL9, had a trend of negative association with INFg, and a trend of positive association with NLR.

Minor points:

  1. Line 95-97. The sentence sounds strange. Please check.

Response: We thank the reviewer for this notion. We have now removed this sentence as it was unnecessary. 

  1. Table 1. Lack of % in morphology.

Response: We thank the reviewer for pointing out these flaws. We have corrected these faults in the manuscript.  

Reviewer 2 Report

this interesting work proposes to document the prognostic links between the deterioration of body composition (sarcopenia, myosteatosis) and survival in colorectal cancers. It also outlines the physiopathological elements related to inflammation.
I regret that the definition of sarcopenia is limited to myopenia without measuring muscle strength. moreover, we have no information on the ingesta and the level of physical activity of the patients before surgery, as well as on their nutritional care before and after surgery. this is detrimental to the correct prognostic evaluation of these parameters. we also do not know if the patients received anti-cancer treatment before the evaluation and how long the disease has been evolving. 13% of patients were metastatic and had palliative surgery cachexia was not characterized in this work and yet it is essential  

Author Response

Reviewer 2: Comments and Suggestions for Authors

This interesting work proposes to document the prognostic links between the deterioration of body composition (sarcopenia, myosteatosis) and survival in colorectal cancers. It also outlines the physiopathological elements related to inflammation.
I regret that the definition of sarcopenia is limited to myopenia without measuring muscle strength. moreover, we have no information on the ingesta and the level of physical activity of the patients before surgery, as well as on their nutritional care before and after surgery. this is detrimental to the correct prognostic evaluation of these parameters. we also do not know if the patients received anti-cancer treatment before the evaluation and how long the disease has been evolving. 13% of patients were metastatic and had palliative surgery cachexia was not characterized in this work and yet it is essential  

Response: We thank the reviewer for these valuable comments. We acknowledge that there are important limitations in the study, as highlighted by the reviewer. We have now further underlined this in the discussion: “In addition to the relatively small sample size, some other limitations should be acknowledged. The definition of sarcopenia was limited to myopenia without measuring muscle strength, as such data was not available for this cohort. Multiple factors may contribute to myosteatosis and sarcopenia, such as the patient’s lifestyle and diet, nutritional care before and after surgery, and other environmental exposures, in addition to effects of the tumor and inflammation. While we extensively characterized tumors and systemic inflammatory markers, we did not collect diet or lifestyle data, potentially resulting in residual confounding. Muscle density and circulating biomarker data were based on single preoperative measurements, and follow-up of their development was lacking. Our analysis was retrospective, and further larger prospective studies are required to validate the findings.”

In addition to clarifying the limitations, we have added subgroup analyses of stage I-III patients (excluding stage IV patients), and the results of have now been added to manuscript (3.6. Sarcopenia and myosteatosis in stage I-III CRC). These results were quite similar to results with stage I-IV patients. However, in stage I-III CRC, myosteatosis statistically significantly associated with overall survival (multivariable HR 1.81, 95% CI 1.00–3.26, p=0.048), while in stage I-IV patients, this finding did not reach statistical significancec (multivariable HR 1.55, 95% CI 0.96–2.50, p=0.075).

Reviewer 3 Report

The author have evaluated the association between sarcopenia/myosteatosis and multiple inflammatory biomarkers in CRC patients. The topic of sarcopenia/myosteatosis is quite important and many studies have been reported as a risk factor of short- and long-term outcome. The results of this manuscript seems to be important to understand the background of sarcopenia/myosteatosis, however there are some criticisms.

  1. Sarcopenia/myosteatosis are known as a risk factor for long-term survival from the various previous studies. However, I do not understand why the sarcopenia and myosteatosis patients would have better prognosis than the sarcopenic or myosteatosis patients. This discrepancy might come from the relatively low number of patient enrolled to this study, but it should have good reason to explain it.
  2. I feel the Stage IV patient should be excluded from the analysis since the patient with distant metastases would be quite different from the patient without distant metastases, especially the inflammation status. Furthermore, it will be difficult to understand the survival analysis in Figure 3.
  3. The author presents that several clinical backgrounds would be significantly associated with sarcopenia and/or myosteatosis in Table 4 and 5. However, I do not understand whether the clinical background leads the patients to  sarcopenia/myosteatosis. Or, is the relationship only the result of sarcopenia/myosteatosis? I feel these data are very descriptive and does not help that much to understand the status of sarcopenia/myosteatosis. Furthermore, the author presented that any of serum cytokines were associated with sarcopenia/myosteatosis. Then, how does the sarcopenia/myosteatosis occur or created in patients? How can we prevent it?
  4. Minor suggestion; I recommend the author to add the p-value between each groups in Figure 3. It is unclear which group is significantly different from other. 

Author Response

Reviewer 3: Comments and Suggestions for Authors

The author have evaluated the association between sarcopenia/myosteatosis and multiple inflammatory biomarkers in CRC patients. The topic of sarcopenia/myosteatosis is quite important and many studies have been reported as a risk factor of short- and long-term outcome. The results of this manuscript seems to be important to understand the background of sarcopenia/myosteatosis, however there are some criticisms.

  1. Sarcopenia/myosteatosis are known as a risk factor for long-term survival from the various previous studies. However, I do not understand why the sarcopenia and myosteatosis patients would have better prognosis than the sarcopenic or myosteatosis patients. This discrepancy might come from the relatively low number of patient enrolled to this study, but it should have good reason to explain it.

Response: We thank the reviewer for the comment. We agree that the associations between sarcopenia/myosteatosis and survival were less clear in this study as compared to some previous studies, which may be related to the rather small sample size. The survival findings of our study also varied according to the endpoint. It is not clear to us, why patients with concurrent sarcopenia and myosteatosis appeared to have better DFS than those with only one muscle abnormality (either sarcopenia or myosteatosis).  However, patients with concurrent sarcopenia and myosteatosis still had worse OS than patients without muscle abnormalities. According to this comment, we have expanded the discussion of this topic line 333-336: “In some of the previous studies (Dolan et al., 2019; Hopkins et al., 2019; Kroenke et al., 2018), sample size has been higher than in our cohort, which might influence our negative result. In particular, it is not clear, why the patients with concurrent sarcopenia and myosteatosis appeared to have similar DFS as those with neither muscle abnormality.”

  1. I feel the Stage IV patient should be excluded from the analysis since the patient with distant metastases would be quite different from the patient without distant metastases, especially the inflammation status. Furthermore, it will be difficult to understand the survival analysis in Figure 3.

Response: We agree with stage IV patients significantly differing from other patients due to their metastatic disease. Thus, we have conducted subgroup analyses of stage I-III patients (lines 229-255, Supplementary Tables 3-7, Supplementary Figure 2). Results were highly concordant regardless of whether metastatic patients were excluded or included.

We have modified Figure 3 and added p-values indicating difference between groups studied. Patients without sarcopenia and myosteatosis were used as the reference group.

  1. The author presents that several clinical backgrounds would be significantly associated with sarcopenia and/or myosteatosis in Table 4 and 5. However, I do not understand whether the clinical background leads the patients to sarcopenia/myosteatosis. Or, is the relationship only the result of sarcopenia/myosteatosis? I feel these data are very descriptive and does not help that much to understand the status of sarcopenia/myosteatosis. Furthermore, the author presented that any of serum cytokines were associated with sarcopenia/myosteatosis. Then, how does the sarcopenia/myosteatosis occur or created in patients? How can we prevent it?

Response: Our results suggest that patients with serrated CRC and severe comorbidities have increased risk of myosteatosis. Serrated CRCs have distinct mutation alterations from conventional CRC. Thus, we hypothesize that our finding may suggest that genetic characteristics of the tumor create cancer-host interaction leading to fat infiltration to muscle. In literature, sarcopenia is associated with many factors, including increased age, as also highlighted in our study.

Molecular mechanism leading to muscle abnormalities that may be targeted are clearly needed to be identified in future studies. We have highlighted this limitation in the discussion: “As the correlations between serum cytokine concentrations and sarcopenia or myosteatosis were generally quite weak, further studies are required to evaluate the molecular mechanisms underlying sarcopenia and myosteatosis.”

  1. Minor suggestion; I recommend the author to add the p-value between each groups in Figure 3. It is unclear which group is significantly different from other.

Response: We agree that one p-value indicating the difference between all 4 study groups does not clearly show the statistical significance of the differences between individual groups. Hence, we used the univariate Cox proportional hazards regression models to compare survival probabilities between study groups, and these p-values are now added to Kaplan-Meier curves. The patients without sarcopenia and myosteatosis were used as the reference group.

Round 2

Reviewer 3 Report

Congratulation to the author for their fine work. They have answered to my questions.